# Applications of Time-Resolved Thermodynamics for Studies on Protein Reactions

**Masahide Terazima**

Department of Chemistry, Graduate School of Science, Kyoto University, Kyoto 606-8502, Japan; mterazima@kuchem.kyoto-u.ac.jp

**Abstract:** Thermodynamics and kinetics are two important scientific fields when studying chemical reactions. Thermodynamics characterize the nature of the material. Kinetics, mostly based on spectroscopy, have been used to determine reaction schemes and identify intermediate species. They are certainly important fields, but they are almost independent. In this review, our attempts to elucidate protein reaction kinetics and mechanisms by monitoring thermodynamic properties, including diffusion in the time domain, are described. The time resolved measurements are performed mostly using the time resolved transient grating (TG) method. The results demonstrate the usefulness and powerfulness of time resolved studies on protein reactions. The advantages and limitations of this TG method are also discussed.

**Keywords:** thermodynamics; kinetics; time resolved; transient grating; protein; reaction

## 1. Introduction

Understanding protein reactions is an important scientific topic. Many thermodynamic and kinetic studies have been performed to determine reaction schemes by detecting reaction intermediates and determining driving force by characterizing the intermediates. Thermodynamic properties, such as molecular volume, enthalpy, entropy, Gibbs energy, and the thermal expansion coefficient, are considered fundamental properties to describe the states. For instance, understanding the molecular recognition processes of ligands and proteins requires the complete characterization of the binding energetics and correlation of thermodynamic data with target structures [1–4]. A quantitative description of the forces that govern molecular associations requires the determination of changes in thermodynamic parameters such as enthalpy change, heat capacity change, and the Gibbs energy of binding. A direct method for measuring heat change during complex formation was developed by isothermal titration calorimetry. Heat is also directly observed by calorimetric methods, such as differential scanning calorimetry. As heat changes occur in almost all chemical and biochemical processes, calorimetry can be used for numerous applications, such as studies on protein folding, and antibody–antigen, protein–protein, enzyme–substrate, and DNA–protein interactions. The informational content of thermodynamic data is large and plays an important role in the elucidation of binding mechanisms.

Another typical experimental methodology for the thermodynamic characterization of reactions is based on the measurements of the temperature ($T$) and pressure ($P$) dependence of the equilibrium constant $K_{eq}$, which requires reaction reversibility [3,4]. The partial molar volume change ($\Delta V$) and enthalpy change ($\Delta H$) of reactions were measured using the following relationships:

$$k_B \left(\partial \ln K_{eq}/\partial(1/T)\right)_P = \Delta H$$
$$-k_B T \left(\partial \ln K_{eq}/\partial P\right)_T = \Delta V \tag{1}$$

where $k_B$ is Boltzmann's constant. These properties are important, because they reflect both the molecular structures and intermolecular interactions.

The translational diffusion of a molecule is driven by the spatial gradient of the chemical potentials of a species and is an important factor for understanding molecular conformation and intermolecular interactions [5]. Measurements of the diffusion coefficient (*D*) using traditional techniques, such as NMR, Taylor dispersion, or capillary methods, require a long time for the measurements. Hence, the transport properties of unstable intermediate biological molecules have not yet been elucidated at all.

Although thermodynamics is a powerful tool for characterizing the molecular state, it is difficult to elucidate the reaction scheme and identify short lived intermediates. On the other hand, kinetics, another important field in science mainly based on spectroscopy—such as optical absorption, emission, or scattering of radiation—has been used to detect reaction intermediates [6]. However, the characterization of intermediates is sometimes difficult, spectroscopically. Therefore, kinetics and thermodynamics can be complementary. If we can measure the thermodynamic properties of transient species during chemical reactions, the accumulated data for stable molecules will enable us to understand these reactions more deeply (Figure 1a). For example, the partial molar volume of a protein is macroscopically observable, and this property is particularly sensitive to hydration and conformation [7–9]. However, little information has been obtained on short lived reaction intermediates. Another example is protein fluctuation, which is an important issue. Proteins fluctuate owing to many local minima of the Gibbs energy surface due to thermal energy. The effect of fluctuations should be considered for a full understanding of protein reactions. The thermodynamics of intermediate species can provide a clue to these dynamics, but this has not yet been accomplished.

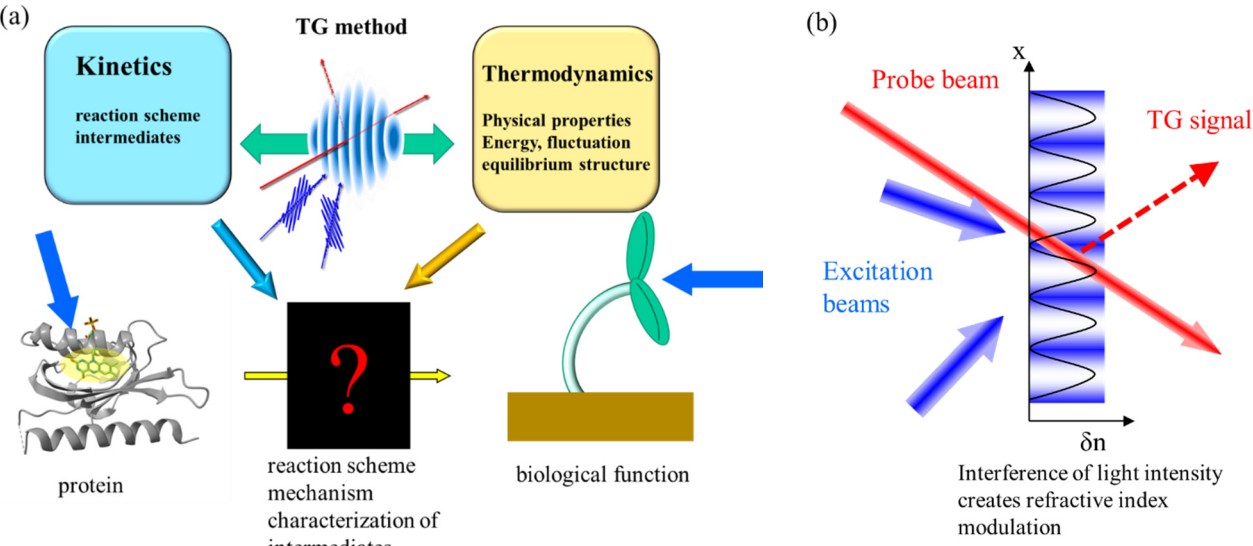

**Figure 1.** (**a**) Schematic illustration showing that the TG method can combine two important fields, kinetics and thermodynamics. Using this method, the reaction scheme, reaction mechanism, and characterization of short lived intermediates, which appear during protein reactions for biological functions, are elucidated. (**b**) Schematic illustration of the TG method. Two excitation beams create an interference pattern of light intensity, and this light produces spatially modulated refractive index change due to various origins. The refractive index change is probed by the diffraction of probe beam.

Herein, our attempts to elucidate protein reaction kinetics and mechanisms by monitoring the thermodynamical properties, including diffusion, are reviewed mostly using the time resolved transient grating (TG) method (Figure 1a).

## 2. Principle

The TG method detects the diffraction of a probe beam using a spatially modulated refractive index created by the interference pattern of two excitation beams (Figure 1b) [10–12].

Upon the photoexcitation of molecules in a condensed phase, a variety of photophysical and photochemical processes occur to change the refractive index ($\delta n$). There are many origins of $\delta n$, such as the temperature elevation (thermal grating, $\delta n_{th}$), density change, volume change (volume grating, $\delta n_v$), and absorption spectrum change (population grating, $\delta n_{pop}$) [13]. The sum of $\delta n_v$ and $\delta n_{pop}$ is referred to as species grating ($\delta n_{spe}$). Under weak diffraction conditions, the TG signal intensity ($I_{TG}$) is approximately proportional to the square of $\delta n$ [14],

$$I_{TG}(t) = \alpha\{\delta n_{th}(t) + \delta n_{spe}(t)\}^2 \tag{2}$$

where $\alpha$ is a constant that represents the sensitivity of the measurement system. From the temporal profile of the signal, the time dependence of enthalpy change ($\Delta H(t)$), volume change ($\Delta H(t)$), diffusion coefficient ($D(t)$), and other thermodynamic properties can be obtained, as described below.

## 3. Results and Discussion

### 3.1. Diffusion Changes

The time profile of $\delta n_{spe}$ reflects the reaction dynamics and molecular diffusion process. If a chemical reaction is completed before the time window of molecular diffusion, the time development of $\delta n_{spe}$ can be expressed as [15,16]:

$$\delta n_{spe}(t) = \delta n_P exp\left(-D_P q^2 t\right) - \delta n_R exp\left(-D_R q^2 t\right) \tag{3}$$

where the subscripts P and R denote the reactant and produced species, respectively, and $q$ is the grating wavenumber. Hence, $D_R$ and $D_P$ are determined simultaneously. Since this measurement can be completed within milliseconds, $D$s of short-lived radicals have been measured and discussed [16].

As the signal observation time can be controlled by choosing $q$, the time dependence of $D_R$ and $D_P$ can be measured. For example, the reaction of

$$R \xrightarrow{h\nu} I \xrightarrow{k} P$$

where R, I, P, and k represent the reactant, intermediate species, final product, and rate constant of the change, respectively, occurring in a two state manner, $\delta n_{spe}(t)$ is given by [17,18]:

$$\delta n_{spe}(t) = \delta n_I exp\left(-(D_I q^2 + k)t\right) + \delta n_P[\tfrac{k}{\{(D_P - D_I)q^2 - k\}}\{exp\left(-(D_I q^2 + k)t\right) - exp\left(-D_P q^2 t\right)\}] - \delta n_R exp\left(-D_R q^2 t\right) \tag{4}$$

where $\delta n_I$ and $D_I$ are the refractive index change and $D$ of the intermediate, respectively. From the time profiles at various values of $q$, $D_R$, $D_I$, $D_P$, and $k$ can be determined.

This method is a powerful and useful approach for studying changes in intermolecular interactions that cannot be detected by other spectroscopic techniques (spectrally silent dynamics). Conformation change that is sensitive to $D$ has been referred as "diffusion-sensitive conformation change (DSCC)". This method has been applied to many protein reactions [19–29]. Two examples are briefly described below.

The first example is the discovery of the reaction intermediates of phototropin (phot), which is a blue light sensor protein that regulates phototropism, chloroplast relocation, and stomatal opening in plants [30,31]. This protein consists of two light-sensing domains (LOV1 and LOV2 domains) and a kinase domain. According to previous studies, which mostly used optical absorption detection, when the LOV domains in the ground state (D-state) are excited to the excited singlet state, a triplet state possessing a broad absorption spectrum (L-state) is produced. In this state, the thiol group of the conserved cysteine covalently binds to the isoalloxazine ring of FMN to yield an adduct state with a blue shifted absorption spectrum (S-state) at a rate of ~μs [32]. The absorption signal only reflects these changes around the chromophore. The reaction dynamics of conformational changes in *Arabidopsis* phot1LOV2 with the linker (phot1LOV2-linker) have been investigated from the

point of view of changes in molecular volume and molecular diffusion by the time resolved TG method [33–39]. Although the absorption spectrum change was completed within a few microseconds, the molecular volume decreased with two time constants, 300 µs and 1.0 ms [35]. The faster phase was attributed to the dissociation step of the linker region from the LOV2 domain ($T^{pre}$-state). After this step, $D$ decreased notably, with a time constant of 1.0 ms from $9.2 \times 10^{-11}$ m$^2$/s to $5.0 \times 10^{-11}$ m$^2$/s, to yield the final product, T-state. This time dependent $D$ was interpreted in terms of the unfolding process of the $\alpha$-helices in the linker region (DSCC). The change in the $\alpha$-helices was confirmed by observing the circular dichroism (CD) intensity. The origin of the $D$ decrease was attributed to an increase in the intermolecular interactions between the protein and water molecules. Based on these observations, a reaction scheme for this protein has been proposed (Figure 2).

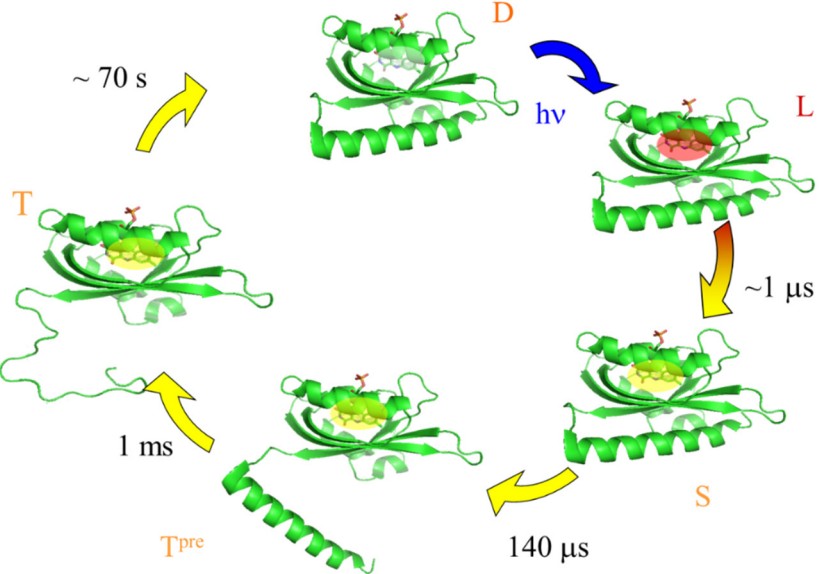

**Figure 2.** Photoreaction scheme of Phot1LOV2-linker.

Another recent example of time resolved diffusion is the reaction scheme for blue light regulated phosphodiesterase 1 (BlrP1) [40]. This is a blue light sensor protein that controls the hydrolysis of cyclic dimeric guanosine monophosphate, which regulates cellular motility and biofilm formation. Changes in the conformation of BlrP1 and its photo-sensing domain, the blue-light-using flavin adenine dinucleotide (BLUF) domain, were investigated using the TG method. Upon photoexcitation, the absorption spectrum shifted to red, but other changes in the BlrP1-BLUF domain were minor, that is, no detectable CD change and no $D$ change. In contrast, a large conformational change in full length BlrP1 was detected as a change in $D$ with a time constant of 21 ms. The extent of the conformational change was concentration dependent, indicating that the dimer form of BlrP1 was responsible for the conformational change. This conformational change was not detected in the CD spectrum, that is, this process is not only absorption spectrum silent but also CD spectrally silent. These features suggested that the observed diffusion change was caused by ternary or quaternary structural changes of the dimer. This conformational change was found to increase significantly with increasing excitation light intensity. This light intensity dependence indicated that DSCC was induced by the photoexcitation of two protomers of the dimer. These results suggest that BlrP1 is not only a photosensor but also a light intensity sensor possessing a nonlinear response to light intensity, and this suggestion was experimentally and directly confirmed by measuring enzymatic activity at various light intensities [41].

### 3.2. Enthalpy and Volume Change

Generally, the photoexcited state of a molecule relaxes to a lower energy state eventually in a condensed phase. A part of the energy is dissipated as emission, into the surrounding matrix by nonradiative transition, or stored as chemical energy by photoreaction. The energy released by the nonradiative transition increases the temperature of the system, and the density of the matrix decreases owing to the thermal expansion. The temperature and density changes give rise to the TG signals termed the temperature grating and density grating, respectively [13]. The sum of the temperature and density gratings is referred to as thermal grating ($\delta n_{th}$).

When the thermal energy from the excited molecule $Q$ is released, $\delta n_{th}$ is given by:

$$\delta n_{th} = \frac{dn}{dT}\frac{Q}{C_p} \tag{5}$$

where $dn/dT$ is the refractive index change due to the temperature variation in the solution and $C_p$ is the heat capacity of the solution. Hence, the amount of released energy, $Q$, can be obtained from the amplitude of $\delta n_{th}$. When the absorbed photon energy ($h\nu$) is released by the nonradiative transition and as fluorescence (emission) photon energy and stored in the molecule (enthalpy change), $\Delta H$ is determined by the following relationship:

$$Q = h\nu - \Phi_f h\nu_f - \Delta H \Phi_r \tag{6}$$

where $\Phi_f$ is the fluorescence quantum yield, $h\nu_f$ is the average photon energy of the fluorescence, and $\Phi_r$ is the quantum yield of the reaction.

The time development of $\delta n_{th}$ represents the rate of nonradiative transition or the heat release process by a chemical reaction. After the creation of the thermal grating component, the signal decayed owing to the thermal diffusion process:

$$I_{TG}(t) = \alpha \left\{\delta n_{th} exp\left(-D_{th}q^2 t\right)\right\}^2 \tag{7}$$

Separating the thermal grating component from the other contributions in a time resolved manner, $\Delta H$ of the reaction is determined from the intensity.

The partial molar volume of a molecule depends on its conformation and intermolecular interactions. For example, the molecular volume changes because of the physical contact between the two molecules, even without the formation of chemical bonds. The volume grating $\delta n_v$ is given by:

$$\delta n_V = \frac{dn}{dV}\Phi_r \Delta V \tag{8}$$

where $dn/dV$ is the density dependence of the refractive index.

The TG method has been applied to many reaction systems, including proteins and relatively small organic molecules [42,43]. For example, this method was applied to $\Delta H$ and $\Delta V$ measurements of the reaction intermediates of the visual pigment octopus rhodopsins [44–46]. Previous absorption measurements revealed that, upon photoexcitation, the photoisomerization of 11-cis retinal to the all trans form and a series of thermal reactions of Batho-Lumi-Meso-AcidMeta occur [47]. The time profile of $\Delta V$ showed that, after the Meso decay, an intermediate (transient Acid-Meta: Tr-Meta) exists before the creation of the AcidMeta. This observation indicates a new intermediate that has been hidden from the absorption detection. $\Delta H$s along the reaction coordinate were determined from the thermal grating signal intensity at various $q$ values (Figure 3) [45].

$\Delta H$ determined by the time resolved method in solution was compared with $\Delta H$, which was previously reported using cryogenic trapping at different low temperatures and traditional calorimetry (Figure 3) [47]. The $\Delta H$ value of Batho at the physiological temperature reasonably agrees with that determined by the cryogenic trapping method (at $-195\,^{\circ}$C). This coincidence suggests that the structural changes in Rh-Batho are localized in a small region around the chromophore, allowing the change even in a frozen

matrix. In the Lumi-Meso process, the enthalpy at physiological temperatures decreased significantly. This change was very different from that determined by the cryogenic trapping method, indicating that the conformation must be different. It should be noted that these enthalpies are for "frozen" structures; peptide backbone and water molecules should be suppressed at a low temperature. This result clearly demonstrates the importance of time resolved measurements under physiological conditions.

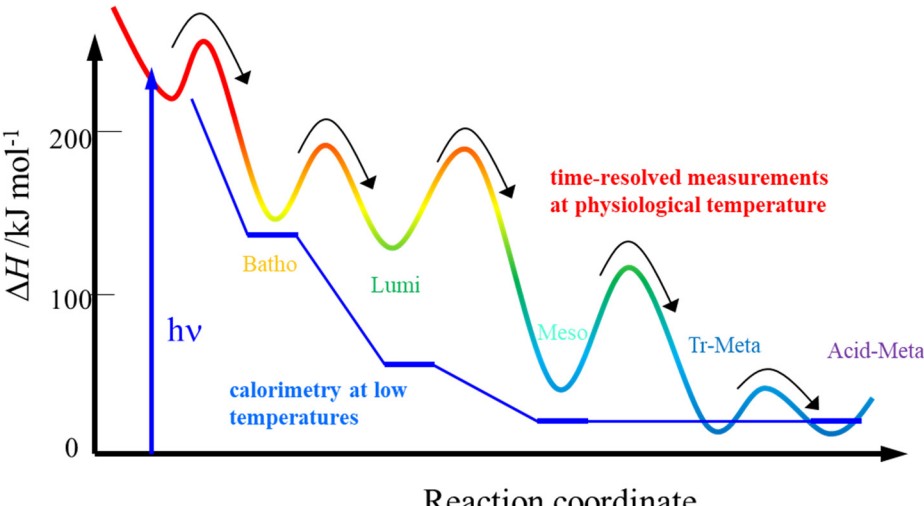

**Figure 3.** Enthalpy changes along reaction coordinates of octopus rhodopsin at 297 K. For comparison, enthalpy changes for trapped intermediates at low temperatures are also shown in blue.

This method has been used to understand the folding process of apoplastocyanin (apoPC) using a photo induced triggering system [48]. ApoPC is a Cu-depleted form of plastocyanin, a small protein (99 amino acid residues) with one small α-helix and eight β-strands. The thermodynamic quantities $\Delta H$, $\Delta V$, and others during the protein folding reaction, were measured in the time domain. Positive $\Delta H$s values were observed during folding. For protein folding, this positive $\Delta H$ must be compensated by positive entropy changes. It has been suggested that the positive entropy change originates from the dehydration effect of hydrophobic residues and/or the translational entropy gain of bulk water molecules [49]. Two phases were observed during the folding processes, and the signs of the volume changes were different. These changes were interpreted in terms of the different relative contributions of the hydration and dehydration of the hydrophilic residues. A two-step hydrophobic collapse in the early stages of folding was suggested.

*3.3. Change in Heat Capacity*

The heat capacity change ($\Delta C_p$) was measured from the temperature dependence of $\Delta H$ using the following equation:

$$\Delta C_p = \left( \frac{\partial \Delta H}{\partial T} \right)_P \tag{9}$$

This method was used to measure the $\Delta C_p$ of some reaction intermediates, such as PYP and phot.

For example, a relatively large thermal expansion volume (0.09 cm³/mol K) and a positive heat capacity change (4.7 kJ/mol K) were detected for the intermediates of LOV2-linker [36]. These characteristic features were interpreted in terms of structural fluctuations and the exposure of hydrophobic residues in the linker domain. Data from this study supports major conformational changes in the linker region in the photochemical reaction of phot, as described in Figure 2.

Heat capacity has been used to characterize the state of noncovalent interactions (e.g., hydrophobic and hydrophilic interactions) of proteins [50]. For example, a positive change in $C_p$ is regarded as a characteristic signature of protein unfolding. The time resolved measurement of $C_p$ is very useful for elucidating the solvation processes of proteins [51].

### 3.4. Thermal Expansion Coefficient Change

The change in the thermal expansion coefficient ($\Delta\alpha_{th}$) was measured from the temperature dependence of the volume change:

$$\Delta\alpha_{th} = (1/V)(\partial\Delta V/\partial T) \tag{10}$$

Hence, the thermal expansion volume $V\Delta\alpha_{th}$ was determined from the slope of the $\Delta V$-$T$ plot.

As a typical example, the temperature dependence of the TG signal after the photoexcitation of PYP has been reported [52]. PYP is a blue light sensor protein initially isolated from the purple sulfur bacterium *Ectothiorhodospira halophila* [53–55]. The chromophore of PYP is 4-hydroxycinnamic acid, which is covalently bound to the Cys side chain. Upon excitation of the chromophore, the ground state (pG) is converted into a red shifted intermediate (pR), which subsequently decays into a blue shifted intermediate (pB) [55]. In addition to the decrease in the thermal grating signal with decreasing temperature, the volume grating intensity also decreased with decreasing temperature. After correcting the temperature dependent reaction yield, $V\Delta\alpha_{th}$ was determined. For example, $\Delta V$ for the pG→pR process was determined to be $-7$ cm$^3$/mol (contraction) at 20 °C, and this contraction increased with decreasing temperature. From the plot of $\Delta V$ vs. $T$, $V\Delta\alpha_{th}$ was determined to be $+0.6$ mL/mol K [52].

It should be noted that the thermal expansion coefficient is related to fluctuations in volume and energy. Many biological systems, such as PYP, are not rigid reaction systems, but there are many local free energy minima (substates) along the reaction coordinate. The observed positive sign of $\Delta\alpha_{th}$ may reflect the structural fluctuation of PYP, and it was suggested that this fluctuation is the driving force of the successive reaction, that is, the unfolding of the N-terminal helices.

### 3.5. Change in Compressibility

The isothermal compressibility change ($\Delta\kappa_T$) is determined by the pressure ($P$) dependence of $\Delta V$ of the reaction.

$$\Delta\kappa_T = -(1/V)\,(\partial\Delta V/\partial p)_T \tag{11}$$

This quantity is particularly important for the detection of fluctuation of biomolecules because the isothermal compressibility ($\kappa_T$) is directly linked to the mean square fluctuations of the protein partial molar volume by:

$$\langle(V - \langle V\rangle)^2\rangle = k_B T\kappa_T \tag{12}$$

where $\langle\ \rangle$ indicates the ensemble average.

Although the relationship between structural fluctuations and reactions is important for elucidating the reaction mechanisms, experimental data describing such fluctuations in reaction intermediates are sparse. To investigate structural fluctuations during a protein reaction, the compressibilities of intermediate species after the photoexcitation of TePixD [56], and the phot1LOV2-linker were measured in the time domain by the TG method [57]. For this measurement, a high pressure cell, which can be used for a small volume of the protein solution, was developed [58]. Using this high pressure cell, it was found that the yield of the S-state formation of phot1LOV2-linker decreased very slightly with increasing pressure. The fraction of reactive species that yields the T-state (linker unfolded state) decreased almost proportionally with the pressure. Compared to the reaction yield, the

volume change was much more sensitive to pressure. Combining these data, the compressibility changes for the short lived intermediate (S-state) and final product (T-state) were determined (Figure 4). The compressibility of the S-state was higher than that of the D-state, and the compressibility decreased from the S-state to the T-state. The change in compressibility was discussed in terms of the cavities inside the protein. Based on the structures in the D- and S-states, the cavity volumes between the LOV domain and the linker domain were found to be increased in the S-state. A larger cavity may lead to larger fluctuations, resulting in the observed enhanced compressibility.

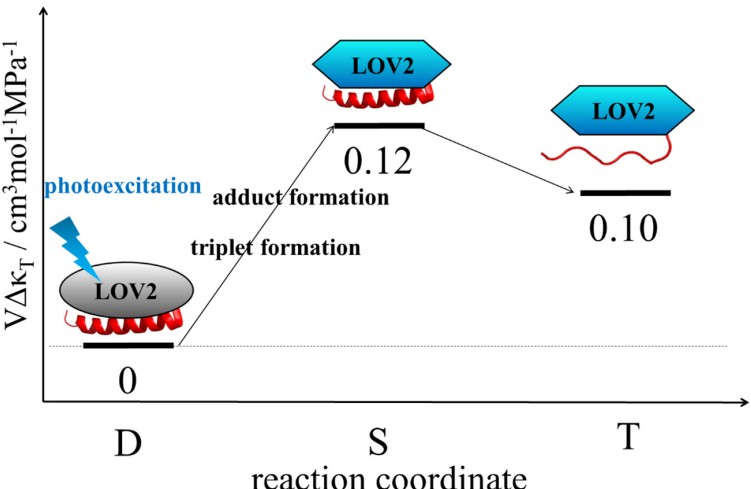

**Figure 4.** Compressibility change in phot1LOV2-linker during photoreaction.

### 4. Conclusions and Perspective

Some applications of the TG method were reviewed to demonstrate the utility and power of studies on protein reaction dynamics. This method can detect $\Delta H(t)$, $\Delta V(t)$, $\Delta \alpha_{th}(t)$, $\Delta \kappa_T(t)$, $\Delta C_p(t)$, and $D(t)$ for short lived transient species in the time domain. Compared with other conventional techniques, such as the temperature dependence of $K_{eq}$ for $\Delta H$, the pressure dependence of $K_{eq}$ for $\Delta V$ measurements, and conventional diffusion detection methods, this method has many significant advantages, as follows.

1.   This method can be used for irreversible reactions, and is also capable of measuring the thermodynamic properties and $D$ in a time resolved manner. Using this method, the $D$ values of various transient, short lived radicals were measured, and it was found that $D$ was very different between stable molecules and transient radicals [15,16].

This method is the only way to study kinetics from the viewpoint of thermodynamics. $\Delta H$ can be measured in a time range of a few tens of picoseconds to a few hundred microseconds. The shortest limit is determined by the acoustic transit time along the grating [59,60], and the longest limit is determined by the thermal diffusion process. To measure $\Delta H$ over a slower time range, another photothermal technique, the transient lens (TrL) method, can be used. In this TrL method, an excitation beam with a spatially Gaussian shape is used instead of the two excitation beams used in the TG method. The light induced change in the refractive index is detected by the spatial change in another probe beam passing through the light-irradiated region. As the thermal diffusion in the illuminated region is of the order of 100 ms, the time development of the thermal energy release can be traced over a much longer time range.

This advantage is particularly important for protein reactions. As there are large degrees of freedom in proteins, it is difficult to elucidate all intermediates only by spectroscopy, and there must be many spectrally silent dynamics. This method can detect spectrally silent dynamics.

2.   This TG technique has a high sensitivity because the homodyne detection of the TG method is background free. Proteins generally and easily form oligomers at

high concentrations, and the reaction is altered by the formation of oligomers. High sensitivity is important for measurements at dilute concentrations. Furthermore, $\Delta V$ of a few cm$^3$/mol was detected. For example, $\Delta V$ of $-7$ cm$^3$/mol obtained for the reaction of PYP was only ~0.7% of the total protein volume [52]. Using a conventional $\Delta V$ measurement, that is, the pressure dependence of $K_{eq}$, $\Delta \kappa_T$ should be calculated from the deviation of the linear relation of $K_{eq}$ vs. *P*. This deviation should be subtle, and it is almost impossible to measure $\Delta \kappa_T$ for any protein solution. Hence, the experimental measurement of this property of a protein in solution is quite difficult. The reported $\Delta \kappa_T$ measurements clearly demonstrate the high sensitivities of $\Delta V$ and $\Delta \kappa_T$. Due to its high sensitivity, a variety of light intensities can be used, and, as a result, the light intensity dependence on the reaction can be studied. Using this merit, some nonlinear light intensity dependence of DSCC has been observed for various photosensor proteins. For example, in addition to BlrP1, shown above, a BLUF protein of decamer PixD (SyPixD) from *Synechocystis* sp. PCC6803 dissociates into five dimers when two protomers in the decamer are excited but not one [61]. Accompanying this dissociation of SyPixD, a cyanobacterial response regulator PixE was released from the PixD$_{10}$-PixE$_5$ complex. This suggests that the biological response of PixD is a nonlinear light intensity sensor for light intensity dependent biological functions.

3. The thermodynamic properties of this method should be exactly along the reaction coordinate. This advantage was demonstrated by the $\Delta H$ measurement of the intermediates of octopus rhodopsin, reported above. Furthermore, this advantage becomes apparent for a case in which the photochemical reaction is pressure dependent. In this case, the reaction scheme can be altered by pressure, and "the molecular volume" cannot be determined from the pressure dependence of $K_{eq}$. However, $\Delta V$ from the volume grating intensity should be an intrinsic value for the reaction. Similarly, the temperature dependence measurement $K_{eq}$ for $\Delta H$ determination could not be correct because protein conformation and the reactivity are sensitive to the temperature. $\Delta H$, which is measured without changing any external properties, should be intrinsic.

4. The TG method does not require any modifications or mutations of the target protein. Some spectroscopic methods require a probe molecule to be attached to the protein. Mutation or binding of a probe molecule may alter the reaction. Hence, the reaction monitored for native proteins using the TG method is appropriate.

5. Water molecules and other nonreactive species in the sample solution do not disturb the measurements. Hence, a variety of sample solutions have been used, such as various pH values, temperatures, solutions containing salts, or crowding macromolecules. The effect of crowding on protein reaction was also examined [62,63]. In some reactions, reactivity was found to be very sensitive to temperature. Based on these findings, it was proposed that this protein may function as a temperature sensor [64].

6. The TG signal originates only from the reacting species. Hence, the presence of nonreacting species does not disturb or interfere with the measurements. This merit is a big advantage compared with other methods that monitor the ensemble average of the reaction system, such as calorimetry, dynamic light scattering, SAXS, and IR spectroscopy. For example, the IR spectrum in the dark state must be subtracted from the light spectrum to obtain any changes in the reaction. If the reaction yield is small, the analysis of the SAXS signal becomes difficult, because of the superposition of the solvent and nonreactive species. Gel chromatography and other diffusion detection methods monitor the ensemble averages of the species in solution. Consequently, it may be difficult to detect oligomer formation in a sample using gel chromatography, unless the population of the dimer is dominant. Moreover, while covalently linked or stable protein aggregates may be detected by a size exclusion chromatography approach, a noncovalent protein aggregate that is formed by weak hydrophobic or hydrogen bond interactions may not be detected because the aggregate might dissociate during elution through the column. The TG technique can overcome these

difficulties, because the measurement is performed in solution, so that even a weak binding complex is maintained in solution.

According to the Stokes–Einstein relationship, $D$ is inversely proportional to the radius of diffusing molecules [5]; therefore, $D$ reflects oligomer formation. In addition, it was found that $D$ is sensitive to the conformation of the proteins [65,66]. $D$ changes if the secondary structure changes, for example, the unfolding or folding of α-helices or β-sheets. It has also been reported that $D$ changes even in tertiary and quaternary protein structures. It should be noted that DSCC is clearly detected even for CD spectrally silent processes. This indicates that the time resolved detection of DSCC is powerful in revealing hidden reaction dynamics.

A shortcoming of this method is that activation of the reactant must be triggered by light. Hence, most of the applications have been reported for photosensory proteins. However, caged compounds can be used to monitor nonphotosensory systems [67,68]. Furthermore, if proteins are labeled with photochromic molecules, $D$ can be measured quickly in a variety of systems that cannot be initiated by light. Recently, the application of the TG method has been expanded to a reaction of nonphotoreactive protein by solution mixing, which is similar to the conventional stopped flow method (stopped flow TG system) [69,70]. In this system, a reaction is initiated by mixing two solutions, and a pulsed grating light is used to monitor the $D$ changes at various time delays from the mixing time. This technique will expand the target systems, not only for photoreactive reactions, but even for more general reactions.

**Funding:** This research received no external funding.

**Data Availability Statement:** Not applicable.

**Acknowledgments:** The author acknowledges all the authors who have contributed to the papers cited in this account.

**Conflicts of Interest:** The author declares no conflict of interest.

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
