# Peer review of "Applications of Time-Resolved Thermodynamics for Studies on Protein Reactions"

_2571-8800, doi:10.3390/j5010014_

Round 1
Reviewer 1 Report
The submitted manuscript is a nice review for the transient grating method developed by the author. The applications described are interesting, and the list of references are useful to see the scope of the method. The present reviewer has only one concern about the meaning of "q" in Eq 3. It is probably (related to) the wavenumber, but the definition of "q" should be spelled out. When this point is corrected, the present reviewer is glad to recommend the acceptance.
Author Response
Author reply> Thank you for this point. I have added the meaning of q (q is the grating wavenumber) below eq.(3).
Reviewer 2 Report
In this manuscript, the author describes his attempts to elucidate protein reaction kinetics and mechanisms by monitoring thermodynamic properties, including diffusion in the time domain from time-resolved measurements.
He reviews some applications of the time-resolved transient grating (TG) method to demonstrate the utility and power of studies on protein reaction dynamics. He notes that this method can detect enthalpy change ΔH(t), partial molar volume change ΔV(t), thermal expansion coefficient change Δαth(t), isothermal compressibility change ΔκT(t), heat capacity change ΔCP(t), and diffusion coefficient D(t) for short-lived transient species in the time domain. The author discusses the advantages and limitations of the TG method.
The author points out that compared with other conventional techniques, such as the temperature dependence of the equilibrium constant Keq for ΔH, the pressure dependence of Keq for ΔV measurements, and conventional diffusion detection methods, this method has many significant advantages:
1) This method can be used for irreversible reactions, and is also capable of measuring the thermodynamic properties and D in a time-resolved manner. It is the only way to study kinetics from the viewpoint of thermodynamics. ΔH can be measured in a time range of a few tens of picoseconds to a few hundred microseconds. This advantage is particularly important for protein reactions, as there are large degrees of freedom in proteins and it is difficult to elucidate all intermediates only by spectroscopy, so there must be many spectrally silent dynamics. The TG method can detect spectrally silent dynamics.
2) The TG technique has a high sensitivity because the homodyne detection of the TG method is background-free. Proteins generally and easily form oligomers at high concentrations, and the reaction is altered by the formation of oligomers. High sensitivity is important for measurements at dilute concentrations. Because of its high sensitivity important for measurements at dilute concentrations, a variety of light intensities can be used, and as a result, the light intensity dependence on the reaction can be studied. Using this merit, some nonlinear light intensity dependence of diffusion-sensitive conformation change (DSCC) has been observed for various photosensor proteins. The authors suggest that the biological response of PixD is a nonlinear light intensity sensor for light intensity-dependent biological functions.
3) The author suggests that the thermodynamic properties of this method should be exactly along the reaction coordinate. He mentions about the demonstration of this advantage by the ΔH measurement of the intermediates of octopus rhodopsin, and that this advantage becomes apparent for the case in which the photochemical reaction is pressure-dependent.
4) The author mentions that the TG method does not require any modifications or mutations of the target protein. Some spectroscopic methods require a probe molecule to be attached to the protein. Mutation or binding of a probe molecule may alter the reaction, and so the reaction monitored for native proteins using the TG method is appropriate.
5) The author observed that as water molecules and other non-reactive species in the sample solution do not disturb the measurements, a variety of sample solutions have been used, such as various pH values, temperatures,
solutions containing salts, or crowding macromolecules. The effect of crowding on protein reaction was also examined. In some reactions, reactivity was found to be very sensitive to temperature. The author mentions that based on these findings, it was proposed in the literature that this protein may function as a temperature sensor.
6) The author discusses that the TG signal originates only from the reacting species, and so the presence of non-reacting species does not disturb or interfere with the measurements. This merit is a big advantage compared with other methods that monitor the ensemble average of the reaction system, such as calorimetry, dynamic light scattering, SAXS, and IR spectroscopy. Because of the superposition of the solvent and non-reactive species, the analysis of the SAXS signal becomes difficult. Gel chromatography and other diffusion detection methods monitor the ensemble averages of the species in solution. Therefore, it may be difficult to detect oligomer formation in a sample using gel chromatography, unless the population of the dimer is dominant. The TG technique can overcome these difficulties, because the measurement is performed in solution, so that even a weak binding complex is maintained in solution.
7) As D is inversely proportional to the 383 radius of diffusing molecules according to the Stokes-Einstein relationship, and so D reflects oligomer formation. The author concludes that D is sensitive to the conformation of the proteins, changes if the secondary structure changes (e.g, unfolding or folding of α-helices or β-sheets), and changes, even in tertiary and quaternary protein structures. He notes that DSCC is clearly detected even for CD spectrally-silent processes, which indicates that the time-resolved detection of DSCC is powerful in revealing hidden reaction dynamics.
The author also discusses that a shortcoming of this method is that activation of the reactant must be triggered by light, and so most of the applications have been reported for photosensory proteins. However, caged compounds can be used to monitor non-photosensory systems, and if proteins are labeled with photochromic molecules, D can be measured quickly in a variety of systems that cannot be initiated by light.
This manuscript presents a comprehensive review of the potential of the new time-resolved transient grating (TG) method to study on protein reaction dynamics. It will be of significant interest for the wide audience of researchers in the area of biomolecular sciences. The manuscript is well written. It deserves publication in J multidisciplinary Scientific Journal in its present form.
Author Response
Author reply> Thank you for the positive comments.